# Weakly-supervised Disentangling with Recurrent Transformations for 3D View Synthesis

**Jimei Yang**[1]    **Scott Reed**[2]    **Ming-Hsuan Yang**[1]    **Honglak Lee**[2]

[1]University of California, Merced
{jyang44, mhyang}@ucmerced.edu
[2]University of Michigan, Ann Arbor
{reedscot, honglak}@umich.edu

## Abstract

An important problem for both graphics and vision is to synthesize novel views of a 3D object from a single image. This is in particular challenging due to the partial observability inherent in projecting a 3D object onto the image space, and the ill-posedness of inferring object shape and pose. However, we can train a neural network to address the problem if we restrict our attention to specific object classes (in our case faces and chairs) for which we can gather ample training data. In this paper, we propose a novel recurrent convolutional encoder-decoder network that is trained end-to-end on the task of rendering rotated objects starting from a single image. The recurrent structure allows our model to capture long-term dependencies along a sequence of transformations, and we demonstrate the quality of its predictions for human faces on the Multi-PIE dataset and for a dataset of 3D chair models, and also show its ability of disentangling latent data factors without using object class labels.

## 1 Introduction

Numerous graphics algorithms have been established to synthesize photorealistic images from 3D models and environmental variables (lighting and viewpoints), commonly known as rendering. At the same time, recent advances in vision algorithms enable computers to gain some form of understanding of objects contained in images, such as classification [16], detection [10], segmentation [18], and caption generation [26], to name a few. These approaches typically aim to deduce abstract representations from raw image pixels. However, it has been a long-standing problem for both graphics and vision to automatically synthesize novel images of applying intrinsic transformations (e.g. 3D rotation and deformation) to the subject of an input image. From an artificial intelligent perspective, this can be viewed as answering questions about object appearance when the view angle or illumination is changed, or some action is taken. These synthesized images may then be perceived by humans in photo editing [14], or evaluated by other machine vision systems, such as the game playing agent with vision-based reinforcement learning [20].

In this paper, we consider the problem of predicting transformed appearances of an object when it is rotated in 3D from a single image. In general this is an ill-posed problem due to the loss of information inherent in projecting a 3D object into the image space. Classic geometry-based approaches either recover a 3D object model from multiple related images, i.e. multi-view stereo and structure-from-motion, or register a single image of a known object class to its prior 3D model, e.g. faces [5]. The resulting mesh can be used to re-render the scene from novel viewpoints. However, having 3D meshes as intermediate representations, these methods are 1) limited to particular object classes, 2) vulnerable to image alignment mistakes and 3) easy to generate artifacts during unseen texture synthesis. To overcome these limitations, we propose a learning approach without explicit 3D model recovery. Having observed rotations of similar 3D objects (e.g. faces, chairs, household objects), the trained model can both 1) better infer the true pose, shape and texture of the object, and 2) make plausible assumptions about potentially ambiguous aspects of appearance in novel

viewpoints. Thus, the learning algorithm relies on mappings between Euclidean image space and underlying nonlinear manifold. In particular, 3D view synthesis can be cast as pose manifold traversal where a desired rotation can be decomposed to a sequence of small steps. A major challenge arises due to the long-term dependency among multiple rotation steps; the key identifying information (e.g. shape, texture) from the original input must be remembered along the entire trajectory. Furthermore, the local rotation at each step must generate the correct result on the data manifold, or subsequent steps will also fail.

Closely related to the image generation task considered in this paper is the problem of 3D invariant recognition, which involves comparing object images from different viewpoints or poses with dramatic changes of appearance. Shepard and Metzler in their mental rotation experiments [22] found that the time taken for humans to match 3D objects from two different views increased proportionally with the angular rotational difference between them. It was as if the humans were rotating their mental images at a steady rate. Inspired by this mental rotation phenomenon, we propose a recurrent convolutional encoder-decoder network with action units to model the process of pose manifold traversal. The network consists of four components: a deep convolutional encoder [16], shared identity units, recurrent pose units with rotation action inputs, and a deep convolutional decoder [8]. Rather than training the network to model a specific rotation sequence, we provide control signals at each time step instructing the model how to move locally along the pose manifold. The rotation sequences can be of varying length. To improve the ease of training, we employed curriculum learning, similar to that used in other sequence prediction problems [27]. Intuitively, the model should learn how to make a one-step $15°$ rotation before learning how to make a series of such rotations.

The main contributions of this work are summarized as follows. First, a novel recurrent convolutional encoder-decoder network is developed for learning to apply out-of-plane rotations to human faces and 3D chair models. Second, the learned model can generate realistic rotation trajectories with a control signal supplied at each step by the user. Third, despite only trained to synthesize images, our model learns discriminative view-invariant features without using class labels. This weakly-supervised disentangling is especially notable with longer-term prediction.

## 2   Related Work

The transforming autoencoder [12] introduces the notion of capsules in deep networks, which tracks both the presence and position of visual features in the input image. These models are shown to be capable of applying affine transformations and 3D rotations to images. We address a similar task of rendering object appearance undergoing 3D rotations, but we use a convolutional network architecture in lieu of capsules (albeit with stride-2 convolution instead of max-pooling), and incorporate action inputs and recurrent structure to handle repeated rotation steps. The Predictive Gating Pyramid (PGP) [19] is developed for time series prediction, and is able to learn image transformations including shifts and rotation over multiple time steps. Our task is related to this time series prediction, but our formulation includes a control signal, uses disentangled latent features, and uses convolutional encoder and decoder networks to model detailed images. Another gating network is proposed in [7] to directly model mental rotation by optimizing transforming distance. Instead of extracting invariant recognition features in one shot, their model learns to perform recognition by exploring a space of relevant transformations. Similarly, our model can explore the space of rotation about an object image by setting the control signal input at each time step of our recurrent network.

The problem of training neural networks that generate images is studied in [25]. A convolutional network mapping shape, pose and transformation labels to images is proposed in [8] for generating chairs. They are able to control these factors of variation and generate high quality renderings. We also generate chair renderings in this paper, but our model adds several additional features: a deep encoder network (so that we can generalize to novel images, rather than only decode), distributed representations for appearance and pose, and recurrent structure for long-term prediction.

Contemporary to our work, the Inverse Graphics Network (IGN) [17] also adds an encoding function to learn graphics codes of images, along with a decoder similar to that in the chair generating network. As in our model, IGN uses a deep convolutional encoder to extract image representations, apply modifications to these, and then re-render. Our model differs in that we train a recurrent network to perform *trajectories* of multiple transformations, we add control signal input at each step, and we use deterministic feed-forward training rather than the variational auto-encoder (VAE) framework [15] (although our approach could be extended to a VAE version).

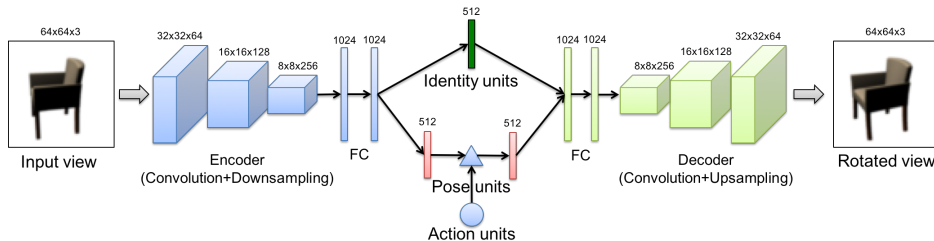

Figure 1: Deep convolutional encoder-decoder network for learning 3d rotation

A related line of work to ours is disentangling the latent factors of variation that generate natural images. Bilinear models for separating style and content are developed in [24], and are shown to be capable of separating handwriting style and character identity, and also separating face identity and pose. The disentangling Boltzmann Machine (disBM) [21] applies this idea to augment the Restricted Boltzmann Machine by partitioning its hidden state into distinct factors of variation and modeling their higher-order interaction. The multi-view perceptron [29] employs a stochastic feedforward network to disentangle the identity and pose factors of face images in order to achieve view-invariant recognition. The encoder network for IGN is also trained to learn a disentangled representation of images by extracting a graphics code for each factor. In [6], the (potentially unknown) latent factors of variation are both discovered and disentangled using a novel hidden unit regularizer. Our work is also loosely related to the "DeepStereo" algorithm [9] that synthesizes novel views of scenes from multiple images using deep convolutional networks.

## 3 Recurrent Convolutional Encoder-Decoder Network

In this section we describe our model formulation. Given an image of 3D object, our goal is to synthesize its rotated views. Inspired by recent success of convolutional networks (CNNs) in mapping images to high-level abstract representations [16] and synthesizing images from graphics codes [8], we base our model on deep convolutional encoder-decoder networks. One example network structure is shown in Figure 1. The encoder network used $5 \times 5$ convolution-relu layers with stride 2 and 2-pixel padding so that the dimension is halved at each convolution layer, followed by two fully-connected layers. In the bottleneck layer, we define a group of units to represent the pose (*pose units*) where the desired transformations can be applied. The other group of units represent what does not change during transformations, named as *identity units*. The decoder network is symmetric to the encoder. To increase dimensionality we use fixed upsampling as in [8]. We found that fixed stride-2 convolution and upsampling worked better than max-pooling and unpooling with switches, because when applying transformations the encoder pooling switches would not in general match the switches produced by the target image. The desired transformations are reflected by the action units. We used a 1-of-3 encoding, in which $[100]$ encoded a clockwise rotation, $[010]$ encoded a no-op, and $[001]$ encoded a counter-clockwise rotation. The triangle indicates a tensor product taking as input the pose units and action units, and producing the transformed pose units. Equivalently, the action unit selects the matrix that transforms the input pose units to the output pose units.

The action units introduce a small linear increment to the pose units, which essentially model the local transformations in the nonlinear pose manifold. However, in order to achieve longer rotation trajectories, if we simply accumulate the linear increments from the action units (e.g. [2 0 0] for two-step clockwise rotation, the pose units will fall off the manifold resulting in bad predictions. To overcome this problem, we generalize the model to a recurrent neural network, which have been shown to capture long-term dependencies for a wide variety of sequence modeling problems. In essence, we turn the pose units to be recurrent to model the step-by-step pose manifold traversals and the identity units are shared across all time steps, since we assume that all training sequences preserve the identity while only changing the pose. Figure 2 shows the unrolled version of our RNN model. We only perform encoding at the first time step, and all transformations are carried out in the latent space; i.e. the model predictions at time step $t$ are not fed into the next time step input. The training objective is based on pixel-wise prediction over all time steps for training sequences:

$$\mathcal{L}_{rnn} = \sum_{i=1}^{N} \sum_{t=1}^{T} ||y^{(i,t)} - g(f_{pose}(x^{(i)}, a^{(i)}, t), f_{id}(x^{(i)}))||_2^2 \tag{1}$$

where $a^{(i)}$ is the sequence of $T$ actions, $f_{id}(x^{(i)})$ produces the identity features invariant to all the time steps, $f_{pose}(x^{(i)}, a^{(i)}, t)$ produces the transformed pose features at time step $t$, $g(\cdot, \cdot)$ is the

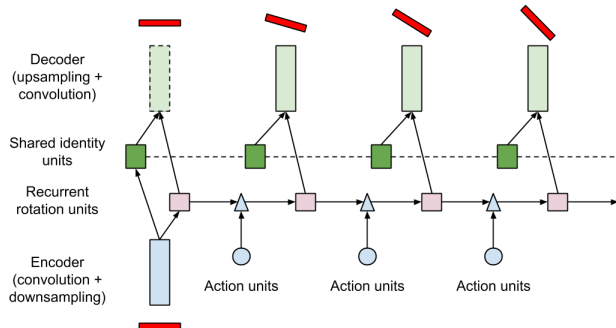

Figure 2: Unrolled recurrent convolutional network for learning to rotate 3d objects. The convolutional encoder and decoders have been abstracted out, represented here as vertical rectangles.

image decoder producing an image given the output of $f_{id}(\cdot)$ and $f_{pose}(\cdot, \cdot, \cdot)$, $x^{(i)}$ is the $i$-th image, $y^{(i,t)}$ is the $i$-th training image target at step $t$.

## 3.1 Curriculum Training

We trained the network parameters using backpropagation through time and the ADAM solver [3]. To effectively train our recurrent network, we found it beneficial to use curriculum learning [4], in which we gradually increase the difficulty of training by increasing the trajectory length. This appears to be useful for sequence prediction with recurrent networks in other domains as well, such as learning to execute Python programs [27]. In section 4, we show that increasing the training sequence length improves both the model's image prediction performance as well as the pose-invariant recognition performance of identity features.

Also, longer training sequences force the identity units to better disentangle themselves from the pose. If the same identity units need to be used to predict both a $15°$-rotated and a $120°$-rotated image during training, these units can not pick up pose-related information. In this way, our model can learn disentangled features (i.e. identity units can do invariant identity recognition but are not informative of pose, and vice-versa) without explicitly regularizing to achieve this effect. We did not find it necessary to use gradient clipping.

## 4 Experiments

We carry out experiments to achieve the following objectives. First, we examine the ability of our model to synthesize high quality images of both face and complex 3D objects (chairs) in a wide range of rotational angles. Second, we evaluate the discriminative performance of disentangled identity units through cross-view object recognition. Third, we demonstrate the ability to generate and rotate novel object classes by interpolating identity units of seen objects.

### 4.1 Datasets

**Multi-PIE.** The Multi-PIE [11] dataset consists of 754,204 face images from 337 people. The images are captured from 15 viewpoints under 20 illumination conditions in different sessions. To evaluate our model for rotating faces, we select a subset of Multi-PIE that covers 7 viewpoints evenly from $-45°$ to $45°$ under neutral illumination. Each face image is aligned through manually annotated landmarks on eyes, nose and mouth corners, and then cropped to $80 \times 60 \times 3$ pixels. We use the images of first 200 people for training and the remaining 137 people as the test set.

**Chairs.** This dataset contains 1393 chair CAD models made public by Aubry et al. [1]. Each chair model is rendered from 31 azimuth angles (with steps of $11°$) and 2 elevation angles ($20°$ and $30°$) at a fixed distance to the virtual camera. We use a subset of 809 chair models in our experiments, which are selected out of 1393 by Dosovitskiy et al. [8] in order to remove near-duplicate models, models differing only in color or low-quality models. We crop the rendered images to have a small border and resize them to a common size of $64 \times 64 \times 3$ pixels. We also prepare their binary masks by subtracting the white background. We use the images of the first 500 models as the training set and the remaining 409 models as the test set.

### 4.2 Network Architectures and Training Details

**Multi-PIE.** The encoder network for Multi-PIE used two convolution-relu layers with stride 2 and 2-pixel padding, followed by one fully-connected layers: $5 \times 5 \times 64 - 5 \times 5 \times 128 - 1024$. The identity

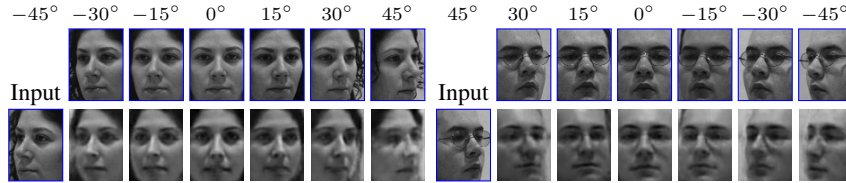

Figure 3: 3D view synthesis on Multi-PIE. For each panel, the top row shows the ground truth images from $-45°$ to $45°$, the bottom row shows the re-renderings of 6-step clockwise rotation from an input image of $-45°$ and of 6-step counter-clockwise rotation from an input image of $45°$.

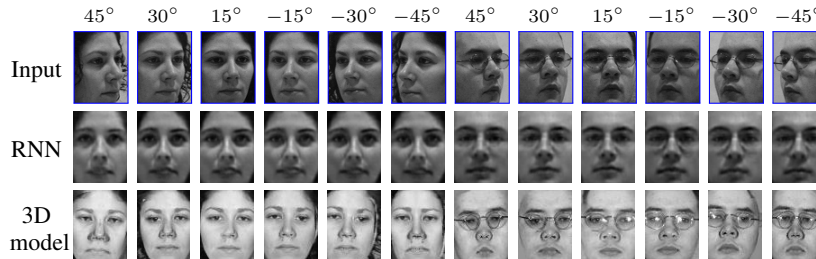

Figure 4: Comparing face pose normalization results with 3D morphable model [28].

and pose units are $512$ and $128$, respectively. The decoder network is symmetric to the encoder. The curriculum training procedure starts with the single-step rotation model which we named RNN1.

We prepare training samples by pairing face images of the same person captured in the same session with adjacent camera viewpoints. For example, $x^{(i)}$ at $-30°$ is mapped to $y^{(i)}$ at $-15°$ with action $a^{(i)} = [001]$; $x^{(i)}$ at $-15°$ is mapped to $y^{(i)}$ at $-30°$ with action $a^{(i)} = [100]$; and $x^{(i)}$ at $-30°$ is mapped to $y^{(i)}$ at $-30°$ with action $a^{(i)} = [010]$. For face images with ending viewpoints $-45°$ and $45°$, only one-way rotation is feasible. We train the network using the ADAM solver with fixed learning rate $1e-4$ for $400$ epochs[1].

Since there are 7 viewpoints per person per session, we schedule the curriculum training with $t=2$, $t=4$ and $t=6$ stages, which we named RNN2, RNN4 and RNN6, respectively. To sample training sequences with fixed length, we allow both clockwise and counter-clockwise rotations. For example, when $t=4$, one input image $x^{(i)}$ at $30°$ is mapped to $(y^{(i,1)}, y^{(i,2)}, y^{(i,3)}, y^{(i,4)})$ with corresponding angles $(45°, 30°, 15°, 0°)$ and action inputs $([001], [100], [100], [100])$. In each stage, we initialize the network parameters with the previous stage and fine-tune the network with fixed learning rate $1e-5$ for 10 additional epochs.

**Chairs.** The encoder network for chairs used three convolution-relu layers with stride 2 and 2-pixel padding, followed by two fully-connected layers: $5\times5\times64 - 5\times5\times128 - 5\times5\times256 - 1024 - 1024$. The decoder network is symmetric, except that after the fully-connected layers it branches into image and mask prediction layers. The mask prediction indicates whether a pixel belongs to foreground or background. We adopted this idea from the generative CNNs [8] and found it beneficial to training efficiency and image synthesis quality. A tradeoff parameter $\lambda = 0.1$ is applied to the mask prediction loss. We train the single-step network parameters with fixed learning rate $1e-4$ for 500 epochs. We schedule the curriculum training with $t=2$, $t=4$, $t=8$ and $t=16$, which we named RNN2, RNN4, RNN8 and RNN16. Note that the curriculum training stops at $t=16$ because we reached the limit of GPU memory. Since the images of each chair model are rendered from 31 viewpoints evenly sampled between $0°$ and $360°$, we can easily prepare training sequences of clockwise or counter-clockwise $t$-step rotations around the circle. Similarly, the network parameters of the current stage is initialized with those of previous stage and fine-tuned with fixed learning rate $1e-5$ for 50 epochs.

### 4.3 3D View Synthesis of Novel Objects

We first examine the re-rendering quality of our RNN models for novel objects instances that were not seen during training. On the Multi-PIE dataset, given one input image from the test set with possible views between $-45°$ to $45°$, the encoder produces identity units and pose units and then the decoder renders images progressively with fixed identity units and action-driven recurrent pose units up to $t$-steps. Examples are shown in Figure 3 of the longest rotations, i.e. clockwise from

$-45°$ to $45°$ and counter-clockwise from $45°$ to $-45°$ with RNN6. High quality renderings are generated with smooth transformations between adjacent views. The characteristics of faces, such as gender, expression, eyes, nose and glasses are also preserved during rotation. We also compare our RNN model with a state-of-the-art 3D morphable model for face pose normalization [28] in Figure 4. It can be observed that our RNN model produces stable renderings while 3D morphable model is sensitive to facial landmark localization. One of the advantages of 3D morphable model is that it preserves facial textures well.

On the chair dataset, we use RNN16 to synthesize 16 rotated views of novel chairs in the test set. Given a chair image of certain view, we define two action sequences; one for progressive clockwise rotation and and another for counter-clockwise rotation. It is a more challenging task compared to rotating faces due to the complex 3D shapes of chairs and the large rotation angles (more than $180°$ after 16-step rotations). Since no previous methods tackle the exact same chair re-rendering problem, we use a k-nearest-neighbor (KNN) method for baseline comparisons. The KNN baseline is implemented as follows. We first extract the CNN features "fc7" from VGG-16 net [23] for all the chair images. For each test chair image, we find its k nearest neighbors in the training set by comparing their "fc7" features. The retrieved top-K images are expected to be similar to the query in terms of both style and pose [2]. Given a desired rotation angle, we synthesize rotated views of the test image by averaging the corresponding rotated views of the retrieved top-K images in the training set at the pixel level. We tune the K value in [1,3,5,7], namely KNN1, KNN3, KNN5 and KNN7 to achieve its best performance. Two examples are shown in Figure 5. In our RNN model, the 3D shapes are well preserved with clear boundaries for all the 16 rotated views from different input, and the appearance changes smoothly between adjacent views with consistent style.

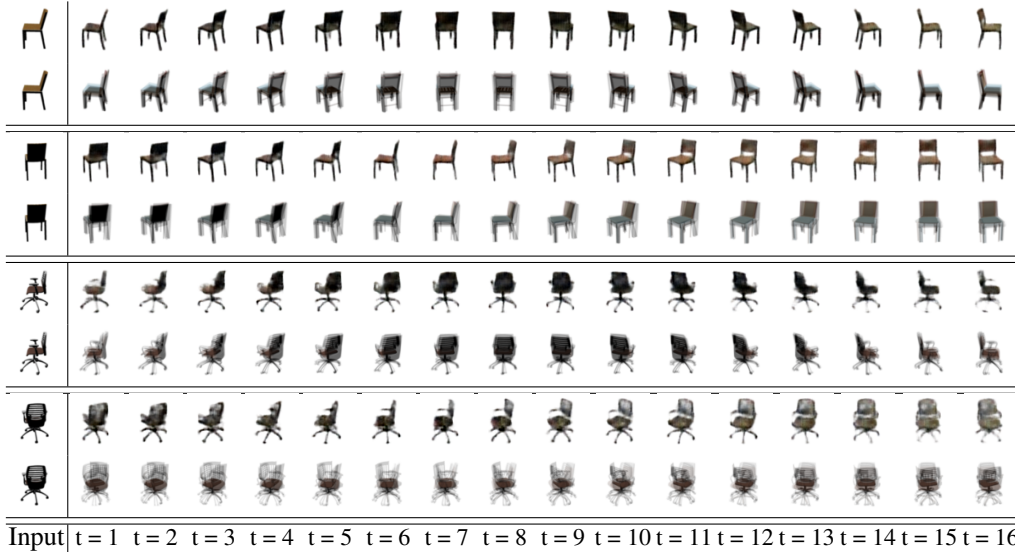

Input | t = 1  t = 2  t = 3  t = 4  t = 5  t = 6  t = 7  t = 8  t = 9  t = 10  t = 11  t = 12  t = 13  t = 14  t = 15  t = 16

Figure 5: 3D view synthesis of 16-step rotations on Chairs. In each panel, the first and second row demonstrate re-renderings of 16-step clockwise and counter-clockwise rotations from our RNN16 model and KNN5 baseline, respectively.

Note that conceptually the learned network parameters during different stages of curriculum training can be used to process an arbitrary number of rotation steps. Unsurprisingly, the RNN1 model (the first row in Figure 6) only works well in the first rotation step and produce degenerate results from the second step. The RNN2 (second row in Figure 6), trained with two-step rotations, generates reasonable results in the third step. Progressively, the RNN4 and RNN8 seem to generalize well on chairs with longer predictions ($t = 6$ for RNN4 and $t = 12$ for RNN8). We measure the quantitative performance of KNN and our RNN by the mean squared error (MSE) in (1) in Figure 7. As a result, the best KNN with 5 retrievals (KNN5) obtains $\sim$310 MSE, which is comparable to our RNN4 model, but significantly outperformed by our RNN16 model ($\sim$179 MSE) with a 42% improvement.

## 4.4 Cross-View Object Recognition

In this experiment, we examine and compare the discriminative performance of disentangled representations through cross-view object recognition.

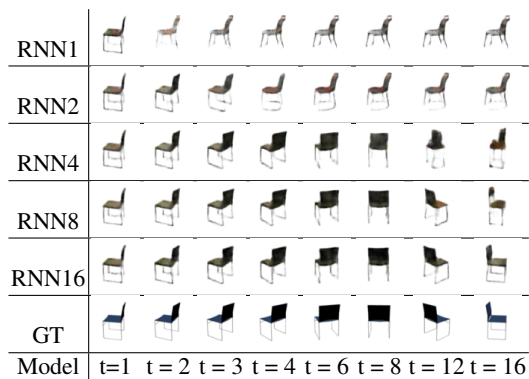

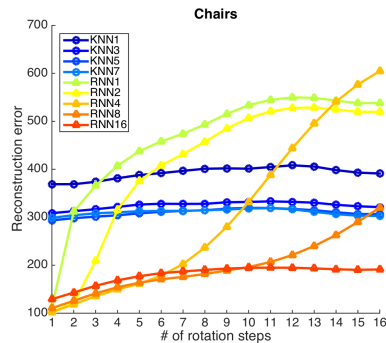

| Model | t=1 | t = 2 | t = 3 | t = 4 | t = 6 | t = 8 | t = 12 | t = 16 |

Figure 6: Comparing chair synthesis results from RNN at different curriculum stages.

Figure 7: Comparing reconstruction mean squared errors (MSE) on chairs with RNNs and KNNs.

**Multi-PIE.** We create 7 gallery/probe splits from the test set. In each split, the face images of same view, e.g. $-45°$ are collected as gallery and the rest of other views as probes. We extract 512-d features from the identity units of RNNs for all the test images so that the probes are matched to the gallery by their cosine distance. It is considered as a success if the matched gallery image has the same identity with one probe. We also categorize the probes in each split by measuring their angle offsets from the gallery. In particular, the angle offsets range from $15°$ to $90°$. The recognition difficulties increase with angle offsets. To demonstrate the discriminative performance of our learned representations, we also implement a convolutional network (CNN) classifier. The CNN architecture is setup by connecting our encoder and identity units with a 200-way softmax output layer, and its parameters are learned on the training set with ground truth class labels. The 512-d features extracted from the layer before the softmax layer are used to perform cross-view object recognition as above. Figure 8 (left) compares the average success rates of RNNs and CNN with their standard deviations over 7 splits for each angle offset. The success rates of RNN1 drop more than $20\%$ from angle offset $15°$ to $90°$. The success rates keep improving in general with curriculum training of RNNs, and the best results are achieved as RNN6. As expected, the performance gap for RNN6 between $15°$ to $90°$ reduces to $10\%$. This phenomenon demonstrates that our RNN model gradually learns pose/viewpoint-invariant representations for 3D face recognition. Without using any class labels, our RNN model achieves competitive results against CNN.

**Chairs.** The experiment setup is similar to Multi-PIE. There are in total 31 azimuth views per chair instance. For each view we create its gallery/probe split so that we have 31 splits. We extract 512-d features from identity units of RNN1, RNN2, RNN4, RNN8 and RNN16. The probes for each split are categorized into 15 angle offsets ranging from $12°$ to $174°$. Note that this experiment is particularly challenging because chair matching is a fine-grained recognition task and chair appearances change significantly with 3D rotations. We also compare our model against CNN, but instead of training CNN from scratch we use the pre-trained VGG-16 net [23] to extract the 4096-d "fc7" features for chair matching. The success rates are shown in Figure 8 (right). The performance drops quickly when the angle offset is greater than $45°$, but the RNN16 significantly improves the overall success rates especially for large angle offsets. We notice that the standard deviations are large around the angle offsets $70°$ to $120°$. This is because some views contain more information about the chair 3D shapes than the other views so that we see performance variations. Interestingly, the performance of VGG-16 net surpasses our RNN model when the angle offset is greater than $120°$. We hypothesize that this phenomenon results from the symmetric structures of most of chairs. The VGG-16 net was trained with mirroring data augmentation to achieve certain symmetric invariance while our RNN model does not explore this structure.

To further demonstrate the disentangling property of our RNN model, we use the pose units extracted from the input images to repeat the above cross-view recognition experiments. The mean success rates are shown in Table 1. It turns out that the better the identity units perform the worse the pose units perform. When the identity units achieve near-perfect recognition on Multi-PIE, the pose units only obtain a mean success rate $1.4\%$, which is close to the random guess $0.5\%$ for 200 classes.

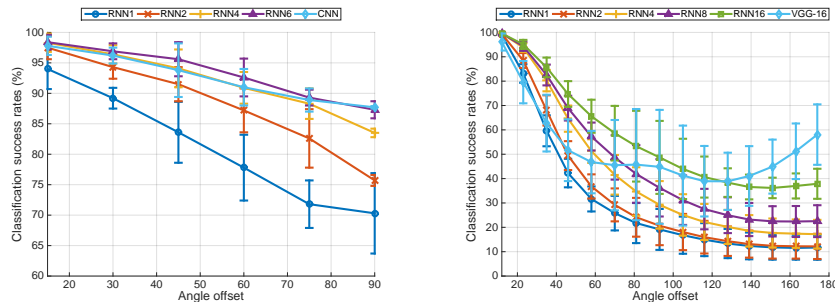

Figure 8: Comparing cross-view recognition success rates for faces (left) and chairs (right).

Table 1: Comparing mean cross-view recognition success rates (%) with identity and pose units.

| Models | RNN: identity | RNN: pose | CNN |
|---|---|---|---|
| Multi-PIE | 93.3 | 1.4 | 92.6 |
| Chairs | 56.8 | 9.0 | 52.5 (VGG-16) |

## 4.5 Class Interpolation and View Synthesis

In this experiment, we demonstrate the ability of our RNN model to generate novel chairs by interpolating between two existing ones. Given two chair images of same view from different instances, the encoder network is used to compute their identity units $z_{id}^1, z_{id}^2$ and pose units $z_{pose}^1, z_{pose}^2$, respectively. The interpolation is computed by $z_{id} = \beta z_{id}^1 + (1-\beta) z_{id}^2$ and $z_{pose} = \beta z_{pose}^1 + (1-\beta) z_{pose}^2$, where $\beta = [0.0, 0.2, 0.4, 0.6, 0.8, 1.0]$. The interpolated $z_{id}$ and $z_{pose}$ are then fed into the recurrent decoder network to render its rotated views. Example interpolations between four chair instances are shown in Figure 9. The Interpolated chairs present smooth stylistic transformations between any pair of input classes (each row in Figure 9), and their unique stylistic characteristics are also well preserved among its rotated views (each column in Figure 9).

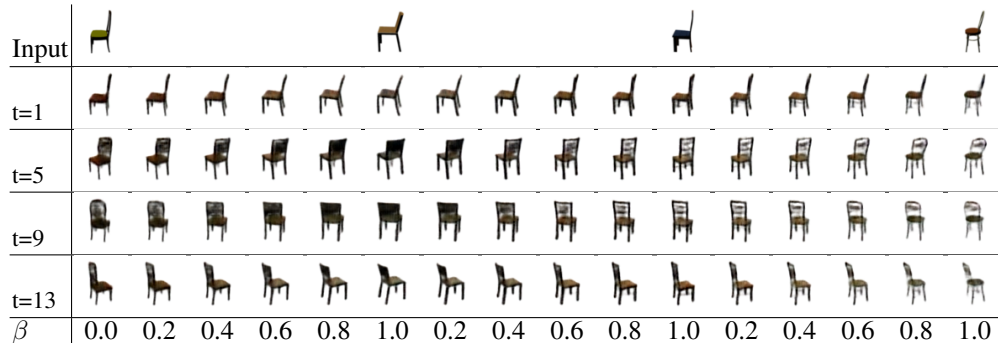

Figure 9: Chair style interpolation and view synthesis. Given four chair images of same view (first row) from test set, each row presents renderings of style manifold traversal with fixed view while each column presents the renderings of pose manifold traversal with fixed interpolated identity.

## 5 Conclusion

In this paper we develop a recurrent convolutional encoder-decoder network, and demonstrate its effectiveness for synthesizing 3D views of unseen object instances. On the Multi-PIE dataset and a database of 3D chair CAD models, the model predicts accurate renderings across trajectories of repeated rotations. The proposed curriculum training by gradually increasing trajectory length of training sequences yields both better image appearance and more discriminative features for pose-invariant recognition. We also show that a trained model could interpolate across the identity manifold of chairs at fixed pose, and traverse the pose manifold while fixing the identity. This generative disentangling of chair identity and pose emerged from our recurrent rotation prediction objective, even though we do not explicitly regularize the hidden units to be disentangled. Our future work includes introducing more actions into the proposed model other than rotation, handling objects embedded in complex scenes, and handling one-to-many mappings for which a transformation yields a multi-modal distribution over future states in the trajectory.

## Footnotes

[1]We carry out experiments using Caffe [13] on Nvidia k40c and Titan X GPUs.

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
