[Reviews · NeurIPS 2015]

Submitted by Assigned_Reviewer_1

In this paper, the authors design a recurrent convolutional encoder-decoder network that can render an object from different pivoting viewpoints given a single 2D image of the object. The authors evaluate the performance of their proposed model via its ability to generate images of faces and chairs from rotated viewpoints. They also perform additional experiments to examine the benefits of curriculum learning, to evaluate the model's performance of disentangled representations through cross-view object recognition, and to explore the model's ability to perform class interpolation with chairs.

Although the proposed model is well presented and detailed, the evaluation of its performance is rather inadequate. The experiments used to demonstrate the effectiveness of the proposed RNN model are largely qualitative (figures with visualizations of results should be scaled to an appropriate size for the reader to see). Introducing a quantitative measure of performance with an appropriate error metric (to compare with state-of-the-art results) should prove to be more insightful than qualitative assessments alone.

In general, the paper is well written and easy to follow. There are a few minor grammatical errors (i.e. line 124 "rotate" -> "rotated").

The significance of this paper is predominantly impaired by its lack of ample experimentation and analysis. It would be relevant to include experiments with more datasets other than Multi-PIE and Chairs. Demonstrating the model's ability to render rotated viewpoints for different types of objects would make the network architecture seem less ad hoc. Additionally, it would also be interesting to see how the proposed model could generalize to handle object rotation trajectories that deviate from a static axis (non-repeating rotations).
Summary: The paper develops a deep learning model for the task of rendering an object from different rotational viewpoints given a single 2D image of that object. Although the approach is novel and interesting, the paper lacks sufficient experimentation and relevant analysis to flesh out the model's significance and potential for results that are comparable to state-of-the-art 'geometry-based approaches' (line 51).

Submitted by Assigned_Reviewer_2

In this paper the authors propose a novel recurrent convolutional encoder-decoder network for learning to apply out-of-plane rotations to 3d objects such as human faces and 3d chair models. The proposed network starts from a basic model, where its encoder network disentangles the input image into identity units and pose units, then with the action units applied on pose units to control the rotation direction, its decoder network which consists of convolution and unsampling decode the identity and pose into an image of rotated object and the corresponding object mask. To support longer rotation trajectories, the proposed network is then extended to have the recurrent architecture where the encoded identity unit of input image is fixed and the pose unit is changed by a sequence of action units, and finally both identity and pose units are fed into decoder to generate the result image.

One of main contribution of this paper is learning to disentangle the representations for identity/appearance and pose factors, where the identity units are shown to be a discriminative view-invariant features in the cross-view object recognition task. In addition, this disentangling properties will benefit more and predict better rendering while using the longer rotation trajectories in the curriculum training stages for training the proposed recurrent convolutional encoder-decoder network.

The paper is well-written, easy to follow, and the motivation for different parts of proposed method is all clearly described. Also the qualitative results for predicted rendering of rotated images and quantitative evaluation on cross-view object recognition task provide good support for the method, especially the disentangled representations for pose and identity factors.

Some minor weakness are listed as follows and hopefully the authors can address them in the rebuttal period:

- The proposed network can only support discrete rotation angles, depending on the set of rotation angles shown in the training data. Do the authors have any initial idea how to extend the proposed method to support continuous rotation angles?

- The proposed recurrent convolutional encoder-decoder network is trained with fixed-length, which is actually contradictory to general recurrent neural networks.

Summary: This paper proposes a novel recurrent convolutional encoder-decoder network that is trained end-to-end on the task of rendering rotated objects starting from a single image. The main contribution of generative disentangling the identity and pose factors which emerged from the recurrent rotation prediction objective is well demonstrated by the qualitative and quantitative evaluations.

Author Feedback
Author rebuttal: We thank reviewers for valuable comments.

R3: Comparison to CNN for cross-view recognition
Our model does not involve discriminative training using identity labels like CNN. This experiment shows that the identity and pose representations can be disentangled by generative training of the proposed recurrent networks. We believe this revealed disentangling property itself is scientifically interesting to the NIPS community.

R1, R3: Objects beyond face and chair; generalization
To our knowledge, previous deep generative models have not demonstrated results on both. Our network architectures consist of simple common operations, e.g. convolution and relu layers without using any category-specific domain-knowledge/designs for faces or chairs. Motivated by your comments, we have also ran experiments on other object categories. For example, our preliminary results on car images are qualitatively similar as those shown in our submission (i.e., our model can perform out-of-plane rotation for unseen car images). We expect similar results will hold for other object categories. We will include these results in the final version.

R1: Comparison to geometry-based approaches
Geometry-based approaches require image alignment to pre-trained 3D models, and are usually highly customized for particular objects, e.g. 3D face morphable models. It is challenging to build such models for common objects, e.g. chairs. In contrast, our model is more generic and does not require significant modification of network architectures when applied to objects of diverse structure (e.g. faces, chairs, and others).

R1, R3: Quantitative comparisons to the state-of-the-art
Since no previous methods tackles the exact same problem as this work, we use a k-nearest-neighbor (KNN) method for baseline comparisons. The KNN baseline is implemented as follows. We first extract the CNN features fc7 from VGGnet for all the chair images. For each query chair image, we find its k nearest neighbors in the training set by comparing their CNN features. The retrieved top-K images should be similar to the query in terms of both style and pose. Given a desired rotation angle, we synthesize the query rotation result by averaging the corresponding rotated views of the retrieved top-K images in the training set at the pixel level. We rotate all the test images up to 16 steps (Fig. 4) and measure the quantitative performance of KNN and our RNN by the mean squared error (MSE) like in Eqn (1). As a result, the best KNN with 5 retrievals (KNN5) obtains ~310 MSE, which is comparable to our RNN4 model (~309 MSE), but it's significantly outperformed by our RNN16 model (~179 MSE ) with a 42% improvement. We will add more detailed results in the final paper.

R1: Better visualization
Due to the space limit, we will present the generated images of actual sizes in the supplementary material and the project webpage.

R3: Real-world applications
Our model can be potentially used for view-invariant 3D object recognition and single-image 3D object reconstruction.

R2: RNN trained with fixed-length?
Our RNN model can be trained with arbitrary length. The fixed-lengths, e.g. 1, 2, 4, 8, 16 are chosen to demonstrate the merits of curriculum training.

R2, R4: Clarifying pose and action units? Support continuous rotation?
The triangle represents a three-way tensor product that takes a old pose vector and an action vector as input and produces a new pose vector. The binary action vector indicates the minimum rotation angle in the training set, so it can be assembled to a sequence for large rotation angles via recurrence. To support continuous rotation angles, we can use "continuous" action vectors. For example, for the 37.5 degree rotation for face images, we can apply two 15 degree rotations first (applying [1, 0, 0] twice), followed by rotation with a fractional action unit [0.5, 0, 0] for the "remainder" angle (7.5 degree).

R4, R6: Regarding mask stream
The mask stream helps regularize the network by providing additional supervision. The final prediction is given by element-wise product of y_{rgb} and y_{mask}. We ran a control experiment that trains the base network in Fig. 1 without the mask stream. It turned out that the base model without the mask steam obtains ~227 MSE while the base model with the mask stream obtains ~117 MSE (~48% improvement) on the test set.

R5: Comparison to Dosovitskiy et al.
Our model is inspired by Dosovitskiy et al; compared to their CNN decoder that generates chair images using full ground-truth factor labels (e.g., identity, pose, etc.), our model does not require labels except for the rotation angles between the image pairs. Furthermore, our model tackles a more challenging problem that takes a single view as input to render rotated views of a 3D object. In fact, the encoder allows our model to perform "zero-shot" rotation of unseen object instances.